# EEG Data Augmentation for Emotion Recognition with a Task-Driven GAN

**Qing Liu, Jianjun Hao * and Yijun Guo**

Beijing Key Laboratory of Network System Architecture and Convergence, Beijing Laboratory of Advanced Information Networks, School of Information and Communication Engineering, Beijing University of Posts and Telecommunications, Beijing 100876, China
* Correspondence: jjhao@bupt.edu.cn

**Abstract:** The high cost of acquiring training data in the field of emotion recognition based on electroencephalogram (EEG) is a problem, making it difficult to establish a high-precision model from EEG signals for emotion recognition tasks. Given the outstanding performance of generative adversarial networks (GANs) in data augmentation in recent years, this paper proposes a task-driven method based on CWGAN to generate high-quality artificial data. The generated data are represented as multi-channel EEG data differential entropy feature maps, and a task network (emotion classifier) is introduced to guide the generator during the adversarial training. The evaluation results show that the proposed method can generate artificial data with clearer classifications and distributions that are more similar to the real data, resulting in obvious improvements in EEG-based emotion recognition tasks.

**Keywords:** data augmentation; electroencephalogram; generative adversarial networks; emotion recognition; task-driven

## 1. Introduction

Emotions are ubiquitous in our daily lives and can affect or even determine our judgments and decisions. For instance, the authors of [1] show that visual complexity plays a positive role in impacting purchase intentions by affecting consumers' feelings of pleasantness and arousal; the findings in [2] suggest that the detection of positive emotion dysfunction could help to identify mild cognitive impairment not related to Alzheimer's disease patients.

In recent years, human–machine communication has become more prevalent due to the popularization of mobile internet and artificial intelligence technology [3], making it more important for machines to have the ability to recognize and respond to human emotions. Emotion recognition usually refers to audio–visual speech recognition [4,5], text emotion recognition [6], physiological signal emotion recognition [7], etc. Compared with other modalities, physiological signals are not influenced by human subjective factors, which can make emotion recognition more objective and reliable. EEG-based emotion recognition has received much attention in recent years due to the rapid development of brain–computer interface technology. EEGs can measure rhythmic oscillations, which contain rich emotional clues; thus, they can provide effective emotional state features.

However, capturing emotional features usually requires a lot of available data due to the complexity of human emotions. Unlike audio and video, which are relatively simple to access, it is difficult to build a large-scale EEG dataset for the high price of EEG equipment. Data augmentation technology (as a promising method used to solve data scarcity issues [8], e.g., geometric transformation, including scaling, horizontal clippings, and rotation) has been widely used in image processing and computer vision tasks. Depth generation models, which are different from the signal level transformation, such as GAN [9], aim to learn the

deep-level representation of real distribution, so that downstream classification tasks can perform better.

Among the current works on GAN-based EEG data augmentation, original EEG signals are generated through the methods proposed in [10,11], which cannot be used directly for emotion recognition. Moreover, the impact on the performance of the downstream classification task was not taken into consideration in the data augmentation process in [10–13], meaning these methods cannot guarantee the classifications of generated samples. Moreover, the network parameters of the GAN structure based on multiple generators [14] increase with the number of sample classifications, making it challenging to expand the modal. Furthermore, none of the works mentioned above retain the spatial information of multi-channel EEG signals during data processing or feature extraction. The method proposed in this paper aims to build a data augmentation network consisting of simple structures for EEG emotion recognition tasks, which can ensure the quality of the artificial data and effectively improve the performance of downstream classification tasks.

## 2. Related Works

### 2.1. Emotion Recognition Based on EEGs

The evaluation of cognitive functions and the status of clinical subjects based on EEGs are important aspects of electronic medical services and in developing a new human–computer interface [15]. One mainstream approach in emotion recognition research involves dividing the recognition process into three steps: data preprocessing, feature extraction, and classification [16,17]. During feature extraction, the goal is to retain as much spatial information and frequency domain information of the data as possible, which is conducive to the subsequent classification work. Some studies designed end-to-end networks to directly process raw EEG data [18], but training such networks usually requires large amounts of data volume and computing resources. Therefore, in this study, we use the extracted features directly rather than the original EEG signal as the augmentation target. At present, the most common EEG datasets (such as SEED [19] and DEAP [20]) are multi-channel EEGs, and these channels generally contain rich spatial information. Therefore, the two-dimensional feature matrix (according to the spatial distribution of EEG channels) is used as the basic structure of EEG features in many studies. In this paper, we extract the feature matrix and design the structure of the data augmentation network accordingly.

### 2.2. Data Augmentation Based on GAN

#### 2.2.1. GAN

The components of the original GAN include the generator and discriminator. In order to generate high-quality artificial data, they compete with (and promote) each other until reaching Nash equilibrium. One major problem is the mode collapse caused by the instability of the discriminator during training. Wasserstein GAN (WGAN) [21] and its improved version [22] proposed by Arjovsky et al. made great progress in training stability. WGAN replaces the JS divergence in the original GAN with the Wasserstein distance between the distributions used to solve the gradient disappearance problem of the generator. In addition, the generation of the original GAN can only fit the data distribution of the whole training set, and there is no limitation on the specific labels of the generated data. However, conditional GANs [23] add labels as additional information input to constrain the specific fitting range of the generated data. By combining these two GAN variants [22,23], CWGAN can generate artificial data with specific tags while ensuring training stability.

#### 2.2.2. EEG Data Augmentation Based on GAN

Hartmann et al. generated original EEG signals with GAN for the first [10]. They improved WGAN-GP to ensure the performance of the network when Pr and Pf were similar, so as to generate single-channel EEG data that were close to the real data in both time and frequency domains. Luo et al. introduced data augmentation based on

CWGAN to the field of EEG-based emotion recognition for the first time [12]. In their work, the distribution of the differential entropy (DE) feature of real EEG signals was learned by CWGAN to generate DE featured with labels to conduct data augmentation on the EEG dataset. The generated EEG features (DE) were evaluated by discriminator loss, maximum mean error (MMD), and two-dimensional mapping. High-quality data after the evaluation were added to the original dataset. Finally, a support vector machine (SVM) was introduced to evaluate the improvement of emotion recognition performance after data augmentation. In the latter study [13], Luo et al. augmented the EEG data of SEED [19] and DEAP [20] by using the generation model based on GAN and VAE, respectively, and verified the classification performance by using SVM and the deep neural network, respectively. Bouallegue et al. designed a generation network based on DCWGAN [24] to generate raw EEG data [11], and used MLP, SVM, and KNN to evaluate the algorithm performance. Regarding CWGAN, Zhang et al. used multiple input generators to obtain more feature information so as to generate more diverse and more categorical EEG features (DE features) [14]. The generator part consists of multiple input layers and a parameter-sharing layer. Each input layer corresponds to one EEG label category (for example, the SEED dataset labels used in this study are divided into three classifications, so three input layers are required). Each input layer combines random noise with multiple generators of various labels to generate data with clear classification, which is conducive to improving the performance of downstream classification tasks. However, under the multi-generator structure, the number of input layers will increase with the increase in the number of data label classifications, resulting in a growing network scale.

Inspired by research in the field of computer vision [25,26], we propose a task-driven CWGAN that incorporates a classifier during generative adversarial training to generate data with clear classifications. At the same time, compared with the structures of multiple generators, this is easier to expand for situations involving multiple classifications.

The main contributions of this paper are as follows:

1.  The multi-channel EEG differential entropy feature matrix, which retains the spatial distribution information of channels, was extracted as the data augmentation target.
2.  For emotion recognition research, this is conducive to the subsequent classification work, i.e., to retain as much spatial information of the data as possible in the feature extraction process. In this study, the augmentation target is the multi-channel EEG differential entropy feature matrix that retains the channel spatial distribution information rather than the original EEG signal.
3.  A task-driven data augmentation method for emotion recognition based on GAN is proposed.
4.  We introduce the classifier into the EEG data augmentation network based on GAN. Both the discriminator and the classifier will provide gradients for parameter optimization. The former promotes the generation of realistic EEG data, and the latter ensures that the generated data will help to classify the performance.
5.  We evaluate the performance of the proposed method from two aspects.
6.  The Wasserstein distance, MMD (maximum mean difference), and reduced dimension visualization (UMAP) methods are used to evaluate the data quality. Four different classifiers, including SVM and Lenet, are used to enhance the accuracy of downstream classification tasks by enhancing the emotional classification evaluation data.

## 3. The Proposed Method

### 3.1. The Basic Structure and Variants of GAN

The principle of GAN involves a minimum–maximum zero-sum game. The players in the game are networks called generators and discriminators, which are usually deep neural networks. Figure 1 shows the basic structure of the original GAN.

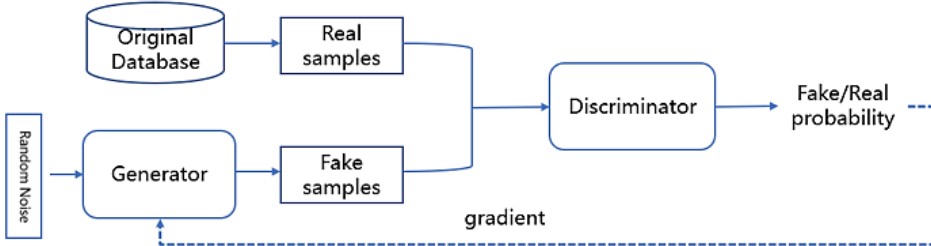

**Figure 1.** The basic structure of GAN.

The function of the generator is to generate artificial samples in the potential space using random Gaussian noise, while the function of the discriminator is to judge whether the given sample is a real sample from the original database or an artificial sample from the generator. The resource of a given sample can be shown from the discriminator output, which represents the probability that the given sample is from the original dataset (namely a true sample).

During the training stage, the parameters of the generator and discriminator networks are adjusted. The discriminator will be more capable of distinguishing between true and false samples, and provide an optimization gradient for the generator so that the generator can generate artificial samples that have closer distributions to those of real samples. Formula (1) shows the objective function of the original GAN:

$$\min_{G} \max_{D} L(X_r, X_g) = \mathbb{E}_{x_r \sim X_r}[\log(D(x_r))] + \mathbb{E}_{x_g \sim X_g}[\log(1 - D(x_g))] \tag{1}$$

There are many variants of GAN, including the principles of WGAN and CGAN mentioned in Section 2.2.1, which can be easily transplanted to various network structures based on GAN, and are widely used in data augmentation.

- WGAN: A major problem of the original GAN is its instability during generative adversarial training resulting from gradient disappearance caused by the discontinuity of the Jensen–Shannon divergence. WGAN formalizes adversarial training by minimizing the Wasserstein distance rather than the Jensen–Shannon divergence between the distribution of generated data and real data so that it can continuously provide useful gradients for the parameter optimization of the generator. Formula (2) shows the objective function of the WGAN:

$$\min_{G} \max_{D} L(X_r, X_g) = \mathbb{E}_{x_r \sim X_r}[D(x_r)] - \mathbb{E}_{x_g \sim X_g}[D(x_g)] \tag{2}$$

- CWGAN: CGAN specifies the types of generated data for the generation network by connecting labels and random noise at the input, so as to ensure the generation of data within a specific range. It can be simply combined with WGAN to form CWGAN. Formula (3) shows the objective function of the CWGAN:

$$\min_{G} \max_{D} L(X_r, X_g, Y) = \mathbb{E}_{x_r \sim X_r, y \sim Y}[D(x_r|y)] - \mathbb{E}_{x_g \sim X_g, y \sim Y}[D(x_g|y)] \tag{3}$$

The proposed method is designed based on the CWGAN.

### 3.2. EEG Data Augmentation for Emotion Recognition

3.2.1. Structure for Multi-Channel EEG Feature Map

As mentioned in Section 2.1, in this paper, we extract the two-dimensional feature map based on the spatial distribution of EEG channels and design the data augmentation network accordingly. In view of the superior performance of the EEG differential entropy features in the EEG-based emotion recognition tasks, we cited the preprocessing and feature extraction methods in [16] for EEG data to obtain the input of the proposed method. Firstly, the original EEG signal is decomposed into four frequency bands. Next, the differential

entropy feature of EEG data from each frequency band is extracted and arranged into a one-dimensional vector. The feature vector is then mapped into a two-dimensional feature matrix according to the spatial distribution of EEG channels in the process of EEG acquisition. Finally, a feature cube of $4 \times 9 \times 9$ is formed by stacking the two-dimensional feature maps of four frequency bands.

For such a structure, we designed a generative adversarial network with classifier-assisted training. Table 1 shows the specific structures of the generator, discriminator, and classifier.

**Table 1.** The structures of the generator (a), discriminator (b), and classifier (c) in the proposed method.

| (a) | |
| --- | --- |
| Layer (type) | Output Shape |
| Linear-1 | [64, 324] |
| ReLU-2 | [64, 4, 9, 9] |
| Conv2d-3 | [64, 64, 9, 9] |
| ReLU-4 | [64, 64, 9, 9] |
| Conv2d-5 | [64, 32, 9, 9] |
| ReLU-6 | [64, 32, 9, 9] |
| Conv2d-7 | [64, 16, 9, 9] |
| Conv2d-8 | [64, 4, 9, 9] |
| **(b)** | |
| Layer (type) | Output Shape |
| Conv2d-1 | [64, 32, 9, 9] |
| LeakReLU-2 | [64, 32, 9, 9] |
| MaxPool2d-3 | [64, 32, 4, 4] |
| Conv2d-4 | [64, 64, 4, 4] |
| LeakReLU-5 | [64, 64, 4, 4] |
| MaxPool2d-6 | [64, 64, 2, 2] |
| Linear-7 | [64, 256] |
| LeakReLU-8 | [64, 256] |
| Linear-9 | [64, 1] |
| **(c)** | |
| Layer (type) | Output Shape |
| Conv2d-1 | [64, 32, 9, 9] |
| LeakReLU-2 | [64, 32, 9, 9] |
| MaxPool2d-3 | [64, 32, 4, 4] |
| Conv2d-4 | [64, 64, 4, 4] |
| LeakReLU-5 | [64, 64, 4, 4] |
| MaxPool2d-6 | [64, 64, 2, 2] |
| Linear-7 | [64, 256] |
| LeakReLU-8 | [64, 256] |
| Linear-9 | [64, 1] |
| Sigmoid-10 | [64, 1] |

### 3.2.2. The Task-Driven CWGAN

First, we built an EEG data generation network based on GAN, then introduce a classifier suitable for EEG emotion recognition to guide the generator during the generative adversarial training. The generator generates false samples for the discriminator and task network, both of which provide gradients for parameter optimization. The former promotes the generation of realistic EEG data, while the latter promotes the generation of clearer data classification so that the network has task awareness and can generate more effective EEG data for emotion recognition tasks.

The framework of the network in the proposed method is shown in Figure 2. The emotion recognition task network is added to the original CWGAN. The generator generates fake samples for the discriminator and task network, both of which provide gradients for parameter optimization.

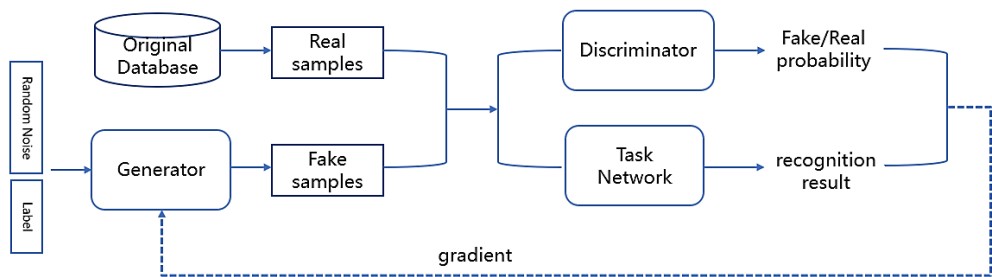

**Figure 2.** The framework of the network based on CWGAN in the proposed method.

The EEG-based data augmentation network designed for emotion recognition tasks takes into account the quality of the generated data (the similarity between generated data and real data) as well as the clarity of generated data classifications so that the performance of emotion recognition tasks on the current dataset can influence the parameter optimization process of the generator. This ensures the improvement of the performance of recognition tasks after data augmentation, and there will be no waste of computing resources.

The network loss function is defined as Formula (4), including the discriminator correlation loss $\min\limits_{G}\max\limits_{D} L(X_r, X_g, Y)$ and classifier correlation loss $\min\limits_{G}\min\limits_{T} L(X_r, X_g, Y)$. The term $\lambda\mathbb{E}_{\widetilde{x}\sim\widetilde{X}, y\sim Y}\left[\left(\left\|\nabla_{\widetilde{x}|y}D(\widetilde{x}|y)\right\|_2 - 1\right)^2\right]$ in Formula (5) (discriminator correlation loss) is a gradient penalty term [22] used to make the training process more stable and the convergence speed faster. The classifier correlation loss is the cross-entropy loss. The coefficient $\rho$ in Formula (4) is used to adjust the proportion of the classifier's correlation loss:

$$L_{G,D,T} = \min\limits_{G}\max\limits_{D} L(X_r, X_g, Y) + \rho \min\limits_{G}\min\limits_{T} L(X_r, X_g, Y) \tag{4}$$

$$\min\limits_{G}\max\limits_{D} L(X_r, X_g, Y) = \mathbb{E}_{x_r\sim X_r, y\sim Y}\left[D(x_r|y)\right] - \mathbb{E}_{x_g\sim X_g, y\sim Y}\left[D(x_g|y)\right]$$
$$-\lambda\mathbb{E}_{\widetilde{x}\sim\widetilde{X}, y\sim Y}\left[\left(\left\|\nabla_{\widetilde{x}|y}D(\widetilde{x}|y)\right\|_2 - 1\right)^2\right] \tag{5}$$

Then the training process is involved. In response to the structure above, the classifier also needs to be trained in each iterative training process of the generator and discriminator, and guide the generator to generate clearer data by providing classification loss-related gradients. The training process is shown in Algorithm 1:

---

**Algorithm 1: Training Process**

---

**Training discriminator and classifier (update the parameters of the discriminator and classifier):**

- Calculate the discriminator loss:
  1:    Place the real sample into the discriminator to obtain the true-or-fake loss of the real sample.
  2:    Place the fake sample generated by the generator into the discriminator to obtain the true-or-fake loss of the fake sample.

- Calculate the classifier loss:
  3:    Place the real sample into the classifier to obtain the classification loss of the real sample.
  4:    Place the fake sample generated by the generator into the classifier to obtain the classification loss of the fake sample.

**Training generator (update the parameters of the generator):**

5:    The random noise is placed into the generating network to generate a fake sample.
6:    The discriminator judges the fake sample to obtain the true-or-fake loss.
7:    The classifier classifies the fake sample to obtain the classification loss.

---

## 4. Experimental Works and Results

### 4.1. Experiment Works

4.1.1. Database

The DEAP dataset [20] is a multimodal dataset collected by Koelstra and others from Queen Mary University in London, as well as other institutions, to study human emotional states. It contains EEG and peripheral physiological signals from 32 participants who viewed 40 music video segments (1 min per segment). The signal was sampled on 48 channels at a frequency of 512 Hz (the first 32 were EEG channels). Moreover, each video was directly scored on a nine-point scale from four dimensions: valence, arousal, dominance, and liking. A pre-processed version of the dataset was provided by the creators for the convenience of users, in which the EEG signals were downsampled to 128 Hz and the EOG was removed. Table 2 presents a pre-processed version of DEAP.

**Table 2.** A summary of the pre-processed version of DEAP; the first three seconds (of each segment) is the resting state of the subject before watching the videos.

| Subjects | Videos | EEG Channels | Sampling Rate | Emotional Dimensions | Label Values |
|---|---|---|---|---|---|
| 32 | 40 | 32 | 128 Hz | Arousal and Valence | Continuous values in the range of 1–9 |
| Data format of each subject (Array shape) | | | | | |
| Videos $\times$ EEG Channels $\times$ Sampling rate $\times$ Segment length = 40 $\times$ 32 $\times$ 128 Hz $\times$ 63 s (3 s in rest and 60 s watching videos) | | | | | |

In this study, the experiment was conducted on two emotional dimensions (valence and arousal) of the multi-channel EEG data in the pre-processed version of DEAP.

4.1.2. Data Processing

The raw EEG signal was subdivided into five different frequency bands in DEAP, on the basis of the intra-band correlation of EEG signals in different human behavioral states. The degree of perception increased with the frequency band, i.e., the high-frequency band was more reflective of the emotional state. At the same time, existing research shows that differential entropy features perform well in EEG emotion recognition [12–14], so we calculated the differential entropy features of four types of high-frequency data reserved in the pre-processed version of DEAP: $\theta$, $\alpha$, $\beta$ and $\gamma$. Differential entropy (DE) is a generalization of Shannon information entropy on continuous variables. Its expression is shown in Formula (6):

$$DE = -\int_a^b p(x)\log(p(x))dx \tag{6}$$

where $p(x)$ is the probability density function of the continuous random variable $x$. The differential entropy of a certain length of an EEG signal obeying the Gauss distribution $N(\mu, \sigma_i^2)$ is defined in Formula (7):

$$
\begin{aligned}
DE &= -\int_a^b \frac{1}{\sqrt{2\pi\sigma_i^2}}\exp\left(-\frac{(x-\mu)^2}{2\sigma_i^2}\right)\log\left(\frac{1}{\sqrt{2\pi\sigma_i^2}}\exp\left(-\frac{(x-\mu)^2}{2\sigma_i^2}\right)\right)dx \\
&= -\frac{1}{2}\log\left(2\pi e\sigma_i^2\right)
\end{aligned}
\tag{7}
$$

The specific band of each sample can be represented by a one-dimensional DE feature vector (length: 32). We used the method in [27] to map the one-dimensional vector into a feature matrix according to the spatial distribution of 32 EEG channels on the EEG electrode map (Figure 3); each non-zero value in the matrix represents the DE value of the corresponding electrode (channel) position. Finally, the feature matrices corresponding to the four frequency bands were stacked to form a $4 \times 9 \times 9$ feature cube. In addition, we divided the emotional scores into high and low categories by five. In this experiment, we conducted the training and testing on two emotional dimensions, i.e., valence and arousal, respectively.

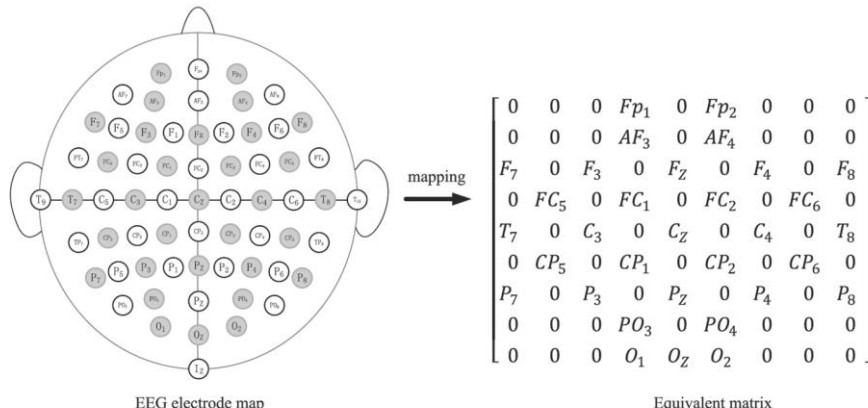

**Figure 3.** Mapping the one-dimensional DE feature vector into a feature matrix according to the spatial distribution of 32 EEG channels.

### 4.1.3. Experiment Details

The setting and adjustment of network parameters, experimental process, and other experimental details will be shown in this section. According to previous relevant studies, the hyperparameter $\lambda$ in Formula (5) was set to 0.1. The grid search was introduced to qualify the hyperparameter $\rho$ in Formula (4) with the measure of minimizing the W-distance between the original dataset and the artificial dataset; finally, 0.1 was chosen as the value of $\rho$. Then, the generative adversarial training was repeated five times to obtain the mean W-distance/MMD. Regarding the evaluation, a five-fold cross-validation was performed to obtain the mean accuracy of the classification task.

### *4.2. Result*

#### 4.2.1. Quality of Generated Data

First, the distribution difference between the original dataset and the artificial dataset, which was generated by the proposed method and the original CWGAN, respectively, was evaluated via two common indicators used to measure the similarity of data distribution, i.e., W-distance and MMD. The abscissa in Figure 4 presents two emotional dimensions, i.e., arousal and valence. The ordinate represents the mean W-distance/MMD value from multiple experiments. The blue and orange bars correspond to the proposed method and the original CWGAN, respectively. The smaller the W-distance or MMD value, the closer it is to the original data distribution. Figure 4 shows that the proposed method improved the distribution similarity compared with the original CWGAN in both emotional dimensions.

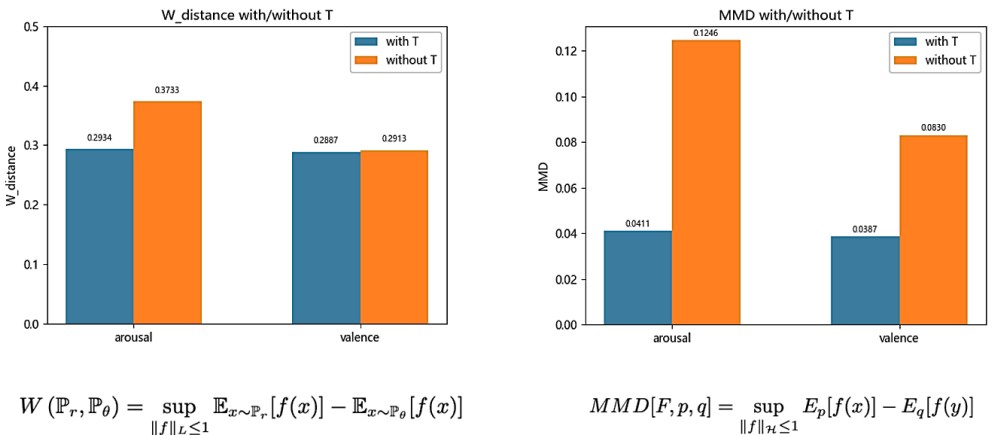

$$W\left(\mathbb{P}_r, \mathbb{P}_\theta\right) = \sup_{\|f\|_L \leq 1} \mathbb{E}_{x \sim \mathbb{P}_r}[f(x)] - \mathbb{E}_{x \sim \mathbb{P}_\theta}[f(x)] \qquad MMD[F, p, q] = \sup_{\|f\|_{\mathcal{H}} \leq 1} E_p[f(x)] - E_q[f(y)]$$

**Figure 4.** W-distance and MMD between the generated data and the original data under the proposed method and the original CWGAN.

Next, UMAP was used to conduct visualization with dimension reduction to observe the distribution of generated data. It can be seen from Figure 5 that the original CWGAN (b) and the proposed method (c) both generate data of corresponding classifications in the original dataset (a); however, the former generates a wide range of banded data, while the latter generates a smaller range of flaky regions besides the original dataset. This is because the classifier introduced a regularization term related to the classification of the generator loss, which limited the generated data to a closer range with the original data of the corresponding classification, indicating that the proposed method makes the generated sample classifications clearer. The three figures on the first line show the EEG DE feature on the emotional dimension (arousal), while the figures on the second line pertain to valence.

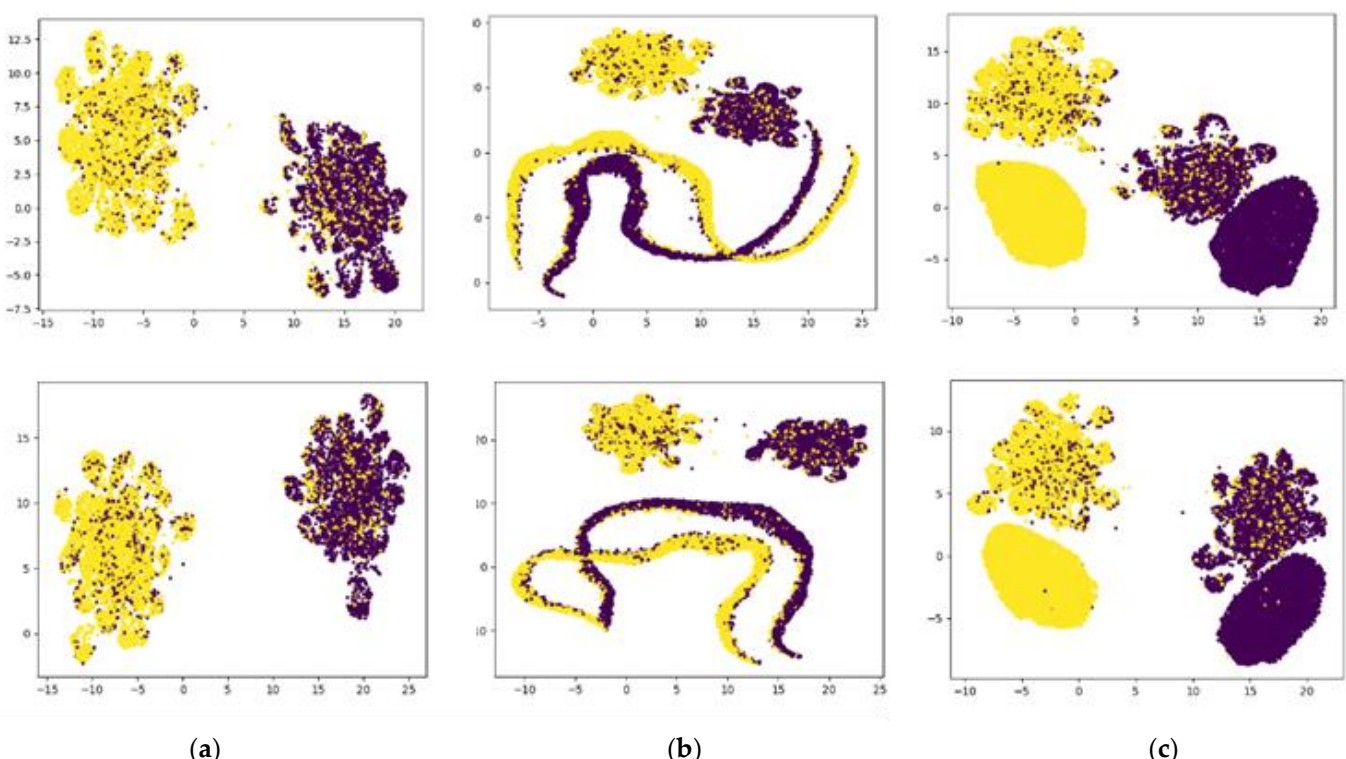

(**a**)　　　　　　　　　　　　　　　(**b**)　　　　　　　　　　　　　　　(**c**)

**Figure 5.** Visualization with UMAP to observe the distribution of the original data (**a**), the data generated by CWGAN (**b**), and the data generated by the proposed method (**c**).

### 4.2.2. Improvement of Performance of Emotion Recognition Task

Finally, we evaluated the improvement in the performance of emotion recognition, mainly focusing on the improvement of classification task accuracy on the testing set by data augmentation using the original CWGAN and the proposed method. Table 3 shows the mean accuracy of emotion recognition in five-fold cross-validation without data augmentation, with data augmentation through the original CWGAN, and with data augmentation through the proposed method, respectively. In this part, SVM and three different neural networks were selected as classifiers to classify emotions in two emotional dimensions: arousal and valence. It can be seen that compared with the original CWGAN, the proposed method (the CWGAN-T in Table 3) can effectively improve the accuracy of classification tasks; the accuracy in the two emotional dimensions improved by 1.5~5.5% and 2~5.2%, respectively.

**Table 3.** The mean accuracy of emotion recognition without data augmentation, with data augmentation through the original CWGAN, and with data augmentation through the proposed method (SVM: support vector machine is often used for emotion recognition of physiological signals; Tasknet: the classifier in the proposed structure; CNN-noPooling: the 4-layer convolutional neural network proposed in related research [16] on EEG emotion recognition; Lenet5 with BN: Lenet5 with Batch-Norm layers).

| Classifiers / Emotional Dimensions | Arousal | | | Valence | | |
|---|---|---|---|---|---|---|
| | Original | CWGAN | CWGAN-T | Original | CWGAN | CWGAN-T |
| **SVM** | 0.8485 | 0.8589 | 0.9135 | 0.8330 | 0.8545 | 0.9064 |
| **Tasknet** | 0.8015 | 0.8876 | 0.9022 | 0.7880 | 0.8755 | 0.8956 |
| **CNN-noPooling** | 0.8499 | 0.9054 | 0.9225 | 0.8346 | 0.8977 | 0.9187 |
| **Lenet-with-BN** | 0.8748 | 0.9050 | 0.9352 | 0.8664 | 0.9021 | 0.9275 |

## 5. Conclusions

This paper presents a task-driven EEG data augmentation network based on CWGAN, which aims to generate artificial EEG data with clearer classifications. The multi-channel EEG differential entropy feature matrix that retains the spatial distribution information of channels was extracted as the data augmentation target, and the emotion classifier was introduced into the original CWGAN structure. In each iterative training process of the generator and discriminator, the classifier guides the generator to generate data with clearer classifications by providing classification loss-related gradient, so as to improve the quality of generated data and effectively improve the accuracy of emotion recognition task. Finally, the proposed method was evaluated in two aspects, namely, the quality of data and the performance improvement of downstream tasks. The Wasserstein distance, MMD (maximum mean difference), and visualization with reduced dimension (UMAP) were used to evaluate the data quality. The results show that the proposed method improved the distribution similarity compared with the original CWGAN in the emotional dimensions of arousal and valence. It can be observed from the visualization that the classifier's introduction of a regularization item to the generator loss resulted in the cleared classification of the generated samples. Four different classifiers, including SVM and Lenet, were used to evaluate the performances of downstream classification tasks after data augmentation. The results show that compared with the original CWGAN, the proposed method can improve the accuracy of classification tasks more effectively, and the accuracy in the emotional dimensions of arousal and valence improved by 1.5–5.5% and 2–5.2%, respectively. However, some problems from this study will need to be further discussed. For instance, qualifying the hyperparameter $\rho$ in Formula (4) can be improved, and penalty terms related to data diversity can be added to the loss function to prevent the generated data from being too centralized and single. These problems will be addressed in future research.

**Author Contributions:** Conceptualization, Q.L.; Methodology, Q.L.; Software, Q.L.; Validation, Q.L.; Investigation, Q.L., J.H. and Y.G.; Resources, J.H. and Y.G.; Writing—original draft, Q.L.; Writing—review & editing, J.H. and Y.G.; Supervision, J.H.; Project administration, J.H. All authors have read and agreed to the published version of the manuscript.

**Funding:** This research received no external funding.

**Informed Consent Statement:** Informed consent was obtained from all subjects involved in the study.

**Data Availability Statement:** The public dataset used in this experiment can be obtained here: http://www.eecs.qmul.ac.uk/mmv/datasets/deap/ (accessed on 7 February 2023).

**Conflicts of Interest:** The authors declare no conflict of interest.

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
