# Peer review of "EEG Data Augmentation for Emotion Recognition with a Task-Driven GAN"

_algorithms, doi:10.3390/a16020118_

Round 1
Reviewer 1 Report
EEG based emotion classification is an interesting and important topic, particularly the data augmentation of EEG.
The paper has several weaknesses that should be improved before publication can be warranted:
a) the introduction and motivation is too brief in the introduction, just some very general statements are given
b) Related work on EEG is not discussed in detail, just 7 references are put together in a single sentence.
c) EEG data sets are not explained in depth, in particular the feature extraction part is missing
d) The statistical evaluation is not explained, e.g is it a LOSA experiment, or just a single experiment? only single numbers are given , variation measure of the methods are not given, statistical tests are missing, etc etc . too much to discuss ..
Author Response
Dear editor,
We sincerely thank you for handling our letter, and also thank the reviewers for their thoughtful and valuable comments, which have greatly helped us to improve the quality of our work. Please see the attachment.

Reviewer 2 Report
The authors propose a method of data augmentation of data from EGG sensors for emotion recognition tasks. The main advantage of the method is that the generated data is then evaluated in a classification process internal to the data generation process. In this way, the authors ensure that the data retains the spatial distribution information of each channel.
The proposed work is relevant mainly due to the introduction of a classifier within the GAN architecture. However, a better analysis of the results is needed within a section that is discussion of the solution. On the other hand, it is not clear how the parameters of the qualifying stage were adjusted. It remains to delve into this process.
minors
1) The abstract must not have references
2) Page 4. The paragraph of Line 158 to 160 is repeated on line 161 to 163.
3) Page 5. Line 187. ", the feature is"
Author Response

(The authors gave the same response as above.)

Reviewer 3 Report
ABSTRACT
No reference should be put in the abstract
INTRODUCTION
I suggest to better frame the context of the research. For instance, the way emotions affect judgments and decisions can be detailed:
Castiblanco et al., User Engagement Comparison between Advergames and Traditional Advertising Using EEG: Does the User’s Engagement Influence Purchase Intention?
RELATED WORKS
Line 61 - All these reference are about "classification"? Is there anything about "data preprocessing" and "feature extraction"? Please detail.
THE PROPOSED METHOD
Describe the emotions object of interest (arousal, valence, dominion, liking) in the proposed method section.
Are you sure it is "dominion" instead of "dominance"?
EXPERIMENTAL WORKS
How did you choose the 40 music video segments? Which were the criteria?
CONCLUSIONS
This section completely lacks of suggestions for future works.
TYPOS
Check again the manuscript: there are some typos such as "a data augmentation networks" at line 51 and "Figure4" at line 259
Author Response

(The authors gave the same response as above.)

Round 2
Reviewer 1 Report
has been improved and can be published
Reviewer 3 Report
The paper has been adequately improved.